# Wholegrain Consumption and Risk Factors for Cardiorenal Metabolic Diseases in Chile: A Cross-Sectional Analysis of 2016–2017 Health National Survey

**DOI:** 10.3390/nu12092815

**Published:** 2020-09-14

**Authors:** Fabian Lanuza, Raul Zamora-Ros, Nicole Hidalgo-Liberona, Cristina Andrés-Lacueva, Tomás Meroño

**Affiliations:** 1Biomarkers and Nutrimetabolomics Laboratory, Department of Nutrition, Food Sciences and Gastronomy, Food Technology Reference Net (XaRTA), Nutrition and Food Safety Research Institute (INSA), Faculty of Pharmacy and Food Sciences, University of Barcelona, 08028 Barcelona, Spain; f.lanuza89@ub.edu (F.L.); rzamora@idibell.cat (R.Z.-R.); n.hidalgoliberona@ub.edu (N.H.-L.); 2Centro de Epidemiología Cardiovascular y Nutricional (EPICYN), Facultad de Medicina, Universidad de La Frontera, Temuco, La Araucanía 4811230, Chile; 3Unit of Nutrition and Cancer, Cancer Epidemiology Research Programme, Catalan Institute of Oncology, Bellvitge Biomedical Research Institute (IDIBELL), 08028 Barcelona, Spain; 4CIBER de Fragilidad y Envejecimiento Saludable (CIBERFES), Instituto de Salud Carlos III, 08028 Barcelona, Spain

**Keywords:** wholegrain, cardiovascular disease, metabolic syndrome, chronic kidney disease, Latin America

## Abstract

Wholegrain (WG) consumption has been associated with reduced risk factors for cardiorenal metabolic diseases (CRMD). In Latin-America. WG intake is low and scarce studies on this subject have been found. We aimed to evaluate the association between WG consumption and risk factors for CRMD in the 2016–2017 Chilean-National Health Survey. This cross-sectional study included 3110 participants representative of a total population of 11,810,647 subjects > 18 y, not taking insulin and with complete data on CRMD risk factors. Outcomes were metabolic syndrome and its components, albuminuria, and impaired glomerular filtration rate (GFR). WG consumption was categorized as regular (≥every two days), sporadic (≥once a month), and non-consumers. Associations were analyzed by multivariable logistic regressions adjusted for confounders taking into account the complex sample design of the survey. Regular WG consumers showed a lower risk of high blood pressure (OR: 0.61, 95%CI: 0.41–0.91) compared to non-consumers in fully-adjusted models. Although inverse associations were noticed with other metabolic syndrome components and impaired GFR, none was statistically significant. The association between WG and BP remained robust in the sensitivity analysis. In conclusion regular WG consumption was associated with a 39% lower risk of high blood pressure in Chilean adults.

## 1. Introduction

Non-communicable diseases are one of the major causes of overall mortality, from which cardiovascular diseases are the leading cause of death worldwide [1]. Cardiovascular, renal, and metabolic diseases (CRMD) share several risk factors and simulation studies have showed that policies and interventions promoting a healthy lifestyle, including a healthy diet, may significantly reduce the incidence rates of CRMD in the years to come [2]. 

Wholegrain (WG) consumption has been associated with a lower incidence of CRMD in a recent meta-analysis of cohort studies and clinical trials [3]. In randomized clinical trials, WG interventions reduced several risk factors for CRMD, such as blood pressure (BP) and total cholesterol levels [3]. However, to our knowledge, no studies have evaluated these associations in a Latin-American population. 

The Latin American Study of Nutrition and Health (ELANS) showed that WG consumption is scarce and represents approximately 1% of total energy intake [4]. Moreover, in the whole ELANS cohort only 36.3% of the participants consumed WG foods; while in the Chilean sub-cohort, this was even lower, at 27.9% [5]. Among the participants who consumed WG in the ELANS-Chile, dietary WG intake was relatively low with an estimated median of 40 g/day (p25–p75: 20–80 g/day) [5]. Similarly, the previous Chile National Health Survey (2009–2010) showed that only 30.3% of the Chilean population regularly consumes WG foods, while 61.3% never consumed them [6].

The National Survey of Food Consumption in Chile (ENCA) showed that only 5% of the Chilean population followed a healthy diet as defined by the Food Based Dietary Guideline for the Chilean population [7,8]. Briefly, this guideline defines a healthy diet, encouraging the consumption of fruits and vegetables five times/day, of dairy products low in fat and sugar three times/day and of legumes and fish at least two times/week, as well as lowering added-salt consumption and avoiding sugar, candy, sweetened drinks and juices, and fried foods [7]. Previous results from the 2016–2017 Chilean National Health Survey showed that the risk for metabolic syndrome was inversely associated with higher consumption of fruits and vegetables or with moderate-vigorous physical activity among participants with normal weight or obesity, respectively [9]. In their analysis, the authors only analyzed the consumption of fruits/vegetables and of fish/seafood, but, similarly to the national dietary guidelines, they omitted WG consumption. Reynolds et al. [3] identified a linear dose-response relationship showing that a 15 g/day increase of WG intake was associated with a reduction of 7% and 12% of relative risk for incident coronary heart disease and type 2 diabetes, respectively. Therefore, the impact of WG consumption on CRMD risk factors has yet to be determined in a Latin-American population characterized by low levels of intake. The aim of the study was to evaluate the association between WG consumption and risk factors for CRMD in the 2016–2017 Chilean National Health Survey.

## 2. Materials and Methods

### 2.1. Study Sample

The 2016–2017 Chilean National Health Survey was a survey with a stratified multistage sample of non-institutionalized participants over 14 years old from urban and rural regions in Chile. Detailed information about the survey has been described elsewhere [10]. In the present study, from the 6233 participants recruited, 3110 subjects with age ≥ 18 years, with complete data on CRMD risk factors and covariates, and not taking insulin were included in this analysis (Figure 1). All participants signed the written consent form. The Ethics Committee from the Pontificia Universidad Católica (PUC) de Chile approved the study protocol (Number: 16–019).

### 2.2. Dietary Exposure

Dietary assessment was carried out by trained interviewers collecting information on the frequency of consumption of fish and seafood, dairy products, WG products, fruits, vegetables, legumes, sugar-sweetened beverages and culinary fat. These questions were designed by experts based on the Food Based Dietary Guideline (FBDG) for the Chilean population as described in the 2009–2010 Chilean National Health Survey [6,7].

WG consumption was classified in three categories according to the question: how often do you consume any WG cereal, such as WG bread, WG cereal or any other food that contains WG flour? Regular WG consumers were participants who answered, “more than once a day”, “once a day” or “every two days”. Sporadic WG consumers were those who answered, “at least once a week” or “once a month”. Non-consumers were those that reported that they “never consume WG foods”.

The consumption of fruits, vegetables and legumes were selected as dietary covariates because they are also important sources of dietary fiber. Consumption of fruits and vegetables was estimated through the questions: “Typically, how many days a week do you eat fruits?” and “Typically, how many days a week do you eat vegetables or vegetable salad? (do not include legumes or potatoes)”, respectively. The answers in days per week were used as continuous variables. Consumption of legumes was assessed through the question: “How often do you consume any type of legumes, such as beans, lentils, green peas or chickpeas?” They were grouped into three categories: (i) “two or more times per week”, (ii) between “at least once a week” and “between one and three times a month”, and (iii) “less than once a month or never”.

### 2.3. Assessment of Risk Factors for Cardiorenal Metabolic Disease

Waist circumference was measured at the midpoint between the costal margin and the iliac crest. Medication use for diabetes and BP was self-reported. BP was measured three times and the mean was used in the study. Fasting blood samples (at least 8 h fast) were drawn for the evaluation of glucose, hemoglobin A1c (HbA1c), creatinine and plasma lipids by standardized methods in a centralized laboratory as described previously. A morning urine sample was collected at the moment of the visit (7–10 am) to measure the urinary albumin-creatinine ratio (uACR) [10]. Urinary albumin values below the lowest reportable value (<0.003 g/L) were replaced by 0.003 g/L and included in the analysis. Very low-density lipoprotein-cholesterol (VLDL-C) and non-high-density lipoprotein-cholesterol (non-HDL-C) and the triglycerides/HDL-C ratio were calculated. The Chronic Kidney Disease-Epidemiology Collaboration (CKD-EPI) formula was used to estimate the glomerular filtration rate (GFR) [11].

### 2.4. Outcomes

Metabolic syndrome was defined according to the Chilean National Guidelines as having at least three of the following five components: high waist circumference (>90 cm for men and >80 cm for women), low HDL (HDL-C <1.03 mmol/L for men and <1.29 mmol/L for women), hypertriglyceridemia (triglycerides ≥ 1.7 mmol/L), high BP (systolic/diastolic BP > 130/85 mmHG or under BP-lowering treatment) and impaired fasting glucose (IFG, glucose > 5.6 mmol/L or under treatment with antidiabetic drugs) [10]. A GFR < 60 mL/min.1.73 m^2^ was used to define chronic kidney disease (CKD) according to the Kidney Disease Improving Global Outcomes (KDIGO) guidelines [12]. uACR was available in 2432 participants. Albuminuria was categorized as microalbuminuria: uACR between 0.34–3.39 mg/mmol, and macroalbuminuria: uACR > 3.39 mg/mmol according to guidelines [12].

### 2.5. Socio-Demographic and Clinical Covariates

Covariates to be included in the study were selected a priori and were based on previous reports of the 2016–2017 Chilean National Health Survey and in a pathophysiological basis [13,14]. Trained interviewers collected socio-demographic data from all participants including geographical area (urban vs. rural), age (years), sex, education level (<8, 8–12, >13 years of education), body mass index (BMI, <25, 25–30, >30 kg/m^2^), physical activity using the Global Physical Activity Questionnaire (GPAQ, low, moderate, high), tobacco use (current smokers vs. non-smokers and former smokers), frequency of alcohol consumption (never, ≤1 time/month, 2–4 times/month, 2–3 times/week, >3 times/week) and number of glasses of alcohol typically consumed (continuous, 0 for non-drinkers), hypertension (self-reported, medication use or systolic/diastolic BP > 140/90 mmHg), diabetes (self-reported or fasting blood glucose > 7 mmol/L) and previous cardiovascular disease (self-reported acute myocardial infarction, stroke or peripheral artery disease). Less than 1% of the participants showed a BMI < 18.5 kg/m^2^ and they were included within the BMI category < 25.0 kg/m^2^. Alcohol dependence was 0.3% in the Survey and these participants were not excluded from the study.

### 2.6. Statistical Analyses

The complex sampling design of the Chilean National Health Survey including strata, cluster and weights were considered for all statistical analyses. Sampling weights and their instructions for use were provided in the database. Categorical data are presented as percentage (standard error, SE), unless otherwise stated. Continuous variables are shown as mean (SE). Generalized linear models adjusted for age (18–24, 25–44, 45–64, >65 years) and sex were used to compare the general characteristics of the population and the levels of risk factors for CRMD across categories of WG consumption. Among these biochemical variables, the skewed-distributed variables were log-transformed before being entered in the analysis. The association between WG consumption and each metabolic syndrome component, metabolic syndrome and CKD was tested using a multivariable logistic regression. The association between WG consumption and albuminuria was tested by multivariable ordinal regression. Odds Ratios (OR) and 95% CI were calculated using the following modeling strategy. Model 1 was adjusted by age (18–24, 25–44, 45–64, >65 years), sex and geographical area. Model 2 was additionally adjusted for BMI (<25, 25–30, >30 kg/m^2^), education level (<8, 8–12, >13 years of education), tobacco use (current smokers vs. former and non-smokers), frequency of alcohol consumption (never, ≤1 time/month, 2–4 times/month, 2–3 times/week, >3 times/week) and number of glasses of alcohol typically consumed (continuous, 0 for non-drinkers), physical activity (GPAQ, low, moderate, high), diabetes (self-reported or fasting blood glucose >126 mg/dl) and previous cardiovascular disease (self-reported acute myocardial infarction, stroke or peripheral artery disease). For renal outcomes (CKD and albuminuria) the model also included hypertension (self-reported, medication use or systolic/diastolic BP > 140/90 mmHg). Model 3 was further adjusted for consumption of fruits, vegetables (times per week) and legumes (categorical, three levels). Interactions between WG consumption and age (categorical), sex, BMI (categorical), education level (categorical), physical activity (categorical) and tobacco use in relation to the risk factors for CRMD were tested by using a likelihood ratio test based on the models with and without an interaction term. If it was significant, the analyses were divided into subgroups. Sensitivity analysis excluding participants from rural areas, having a BMI < 18.5 kg/m^2^, participants with alcohol dependence, patients with diabetes and previous cardiovascular disease was also conducted. SPSS version 25 (IBM, Armonk, NY, USA) was used for all statistical analyses.

## 3. Results

### 3.1. General Characteristics of the Population According to the Frequency of WG Consumption

The studied population consisted of 3110 participants and after applying sampling weights this population was representative of a total *n* (95%CI) of 11,810,647 (10,982,016–12,639,278) subjects. General characteristics are shown in Table 1. In general, the Chilean population featured an elevated prevalence of overweight/obesity (77.5%), metabolic syndrome (41.7%), and physical inactivity (34.7%). The prevalence of hypertension, diabetes and cardiovascular disease was 27.1%, 12.8% and 9.6%, respectively.

Regular WG consumers were younger, more likely to reside in urban areas, to be women, non-smokers or former smokers and to have a higher education level and physical activity than sporadic and non-WG consumers (Table 1). The consumption of fruits and vegetables per week was higher in regular WG consumers and they typically consumed fewer glasses of alcohol (although they did not consume alcohol more frequently). There were no differences in the prevalence of diabetes or previous cardiovascular disease across categories of WG consumption (Table 1).

### 3.2. Risk Factors for CRMD According to the Frequency of WG Consumption

The prevalence of hypertension, metabolic syndrome and CKD was similar across categories of WG consumption (Table 1).

However, WG consumers showed lower levels of several risk factors for CRMD than sporadic WG consumers and non-consumers in age- and sex-adjusted models (Table 2). In particular, regular WG consumers showed lower systolic/diastolic BP and a less atherogenic lipid profile characterized by higher HDL-C and lower triglycerides and VLDL-C (Table 2). Markers of renal function (creatinine, GFR and uACR) were similar across categories of WG consumption.

### 3.3. Association of WG Consumption with Cardiorenal Metabolic Outcomes

The prevalence of individual metabolic syndrome components in the participants was 76.6% for high waist circumference, 52.2% for low HDL, 37.6% for high BP, 34.8% for hypertriglyceridemia, and 24.9% for IFG.

Age-, sex- and geographical area-adjusted models showed an inverse association of WG consumption with high BP and CKD (Appendix A). Results from the fully adjusted model are shown in Appendix A and Figure 2. Regular WG consumption was significantly associated with a 39% lower risk of high BP than non-consumers.

The interaction between WG consumption and age for association with MetS was the only significant interaction among the other risk factors for CRMD. In consequence, MetS analyses were repeated in each of the age groups. An inverse association was observed in the 18–24 y (OR_regularWG vs. non-consumer_: 0.30, 95%CI: 0.06–1.47), 45–64 y (OR_regularWG vs. non-consumer_: 0.76, 95%CI: 0.43–1.34) and >65 y (OR_regularWG vs. non-consumer_: 0.71, 95%CI: 0.29–1.72) age groups, but not in the group of 25–44 y (OR_regularWG vs. non-consumer_: 1.16, 95%CI: 0.62–2.17). There were no significant interactions between WG consumption and the other investigated covariates (sex, education level, BMI, physical activity, and tobacco use) of the study for all outcomes.

### 3.4. Sensitivity Analysis

Excluding participants from rural areas (*n*: 1,305,651), with underweight (BMI < 18.5 kg/m^2^, *n*: 107,656), with alcohol dependence (*n*: 10,770), with diabetes (*n*: 1,513,041) or previous cardiovascular disease (*n*: 1,137,769) showed similar results for the association between WG consumption and high BP (Appendix A). The rest of the results remained non-significant.

## 4. Discussion

This is the first report to show that regular WG consumption is associated with lower levels of risk factors for CRMD, and especially lower risk of high BP, in a Latin American study. Indeed, regular WG consumers showed approximately a 39% lower risk of presenting high BP than non-consumers in this nationwide study representative of almost 12,000,000 Chilean adults.

The inverse association between regular WG consumption and high BP was robust and was in agreement with other studies [14]. Elevated BP is a known-causal risk factor for cardiovascular disease. In randomized clinical trials, a 20% reduction in cardiovascular events was observed for each 10 mmHg of lower systolic BP using medical therapy [15]. Indeed, patients with slight elevations of BP (normal-high BP: 130–139/80–85 mmHg) showed an increased risk of cardiovascular disease [16], and it was shown that patients with other cardiovascular risk factors may benefit more from lower BP at any given value [17]. Therefore, the effect of regular WG consumption on BP may contribute to the prevention of CRMD. It is important to bear in mind that the inverse association between WG consumption and hypertension was confirmed in prospective studies in the US [18] and Asian populations [19]. Thus, a great deal of evidence suggests that low WG consumption may be a causal risk factor for hypertension.

The inverse association between regular WG consumption and high BP may rely, to some extent, on the anti-inflammatory effects attributed to WG and to one of its major nutrients, dietary fiber [20]. These effects could be mediated by beneficial modification of gut microbiota composition and/or the enhancement of short-chain fatty acid production [21,22,23]. Chronic inflammation, typical of obesity and its metabolic derangements, is a key player in the development and progression of endothelial dysfunction, and consequently hypertension [24]. Metagenomic and metabolomic studies are needed to clarify the exact role of gut microbiota composition or its fermentation products in the BP lowering activities attributed to WG.

In our study, we did not observe a significant association between WG consumption and MetS. In a study of 827 Tehranian adults [14], participants in the highest quartile of WG intake showed a 32% lower risk for MetS than those in the lowest quartile. The population characteristics, which included older participants with a higher prevalence of overweight/obesity, hypertension and diabetes, and the lack of a quantitative analysis of WG consumption might explain the different results. Moreover, in the Tehran study, the mean WG consumption was 93 ± 29 g/day, much higher than the estimated mean consumed in the whole sample of Chilean participants in the ELANS, i.e., ±26 g/day [5]. In fact, in a US study on dietary fiber and its association with MetS, risk estimates showed an inverse linear trend across quintiles of intake [25]. Therefore, the low intake level of WG foods in Chile might explain why we did not observe an association between WG consumption and other CRMD outcomes.

The Nutrition and Chronic Diseases Expert Group (NutriCoDE) [26] showed causal protective cardiorenal metabolic effects of WG foods and established that an intake of 50 g/day of WG was associated with a 12% lower risk of coronary heart disease events due to its effects on BP and LDL-C [26].The optimal WG intake range is still under debate and although a range of 100 to 125 g/day has been proposed [27], another study suggested that a WG intake of 210–225 g/day was needed to reduce the risk of chronic diseases and mortality [28]. The HELGA study, which combined samples from three Scandinavian cohorts with a median WG intake of 121 g/day, showed a 44% and 15% lower cardiovascular mortality for women and men, respectively, in the highest quartile of WG intake compared to the lowest one [29]. However, WG foods can provide several health benefits even at low intake doses. In two large prospective cohort studies from the US with relatively low WG intakes (from quintile 1 to quintile 5 of 4.3 g/day to 35.6 g/day, and of 5.8 g/day to 52.6 g/day, respectively) the highest quintiles of WG intake were associated with a 9% lower cardiovascular mortality independent of other dietary and lifestyle factors [30]. Thus, efforts to promote WG consumption in Chile disregarding the level of intake are expected to impact the incidence of CRMD. The Food Based Dietary Guideline for the Chilean population promulgates 11 main statements to define a healthy diet [7]. However, no message directly alludes to WG. Some probable restraints in the intake of WG foods in the Chilean population have already been mentioned such as lower palatability, unawareness of health benefits, and higher prices compared to refined grain foods [31].

Consumption of WG changes considerably between populations. WG rye products are commonly consumed in Scandinavian countries, while WG bread and breakfast cereals are common in the US, and brown rice in Asian countries [28]. Thus, the type of grain may also have some influence when comparing our results with studies from other geographical regions. In particular, WG oats showed a potent cholesterol-lowering effect compared to other grains [32]. As shown in the National Survey of Food Consumption in Chile (ENCA) Survey, the cereals most consumed in Chile were bread and pasta, while breakfast cereals were seldom consumed. Although, it was not explicitly addressed, WG wheat was the most likely consumed WG among Chilean participants.

This was a cross-sectional study and causal relationships cannot be ascertained. This study was based on the frequency of WG consumption and, therefore, future studies quantifying WG intake and its association with CRMD risk factors in Latin America are warranted. Moreover, energy intake was not available for adjustment. However, the use of a national representative sample with sampling weights to adjust for overrepresentation, selection probability, non-response rates, and the poststratification adjustment, carried out by trained personnel, with centralized laboratory results, are major strengths of the study. Although the statistical analyses were adjusted by sociodemographic covariates, some residual confounding cannot be ruled out. In particular, CKD prevalence was somewhat lower than expected in the sample (2.9%). Indeed, a Chilean study found a prevalence of CKD of 12.1% in primary care health centers [33]. Finally, the Chilean population has one of the highest prevalence of overweight and obesity compared to other countries from the region and the results might not be generalized to the whole region [34]. Nonetheless, this is the first study on the association between WG consumption and CRMD risk factors carried out in a Latin American country and our results may contribute to promote research on the cardiorenal metabolic effects of WG foods in the Region.

## 5. Conclusions

Regular WG consumption was associated with lower levels of risk factors for CRMD and especially lower risk of high BP. The benefits of regular WG consumption on BP were independent of socio-demographic characteristics and other dietary sources of fiber. The adjustment for other dietary fiber sources suggests that additional components within WG may also contribute to its BP-lowering effects [35]. Considering its health benefits, access to and consumption of WG foods should be included and promoted by national dietary guidelines and food policies in order to prevent CRMD in Latin America.

## Figures and Tables

**Figure 1 nutrients-12-02815-f001:**
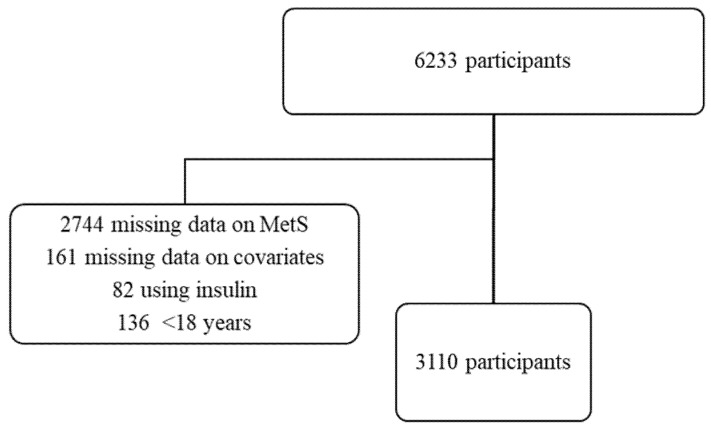
Flowchart of participants in the 2016–2017 Chilean National Health Survey.

**Figure 2 nutrients-12-02815-f002:**
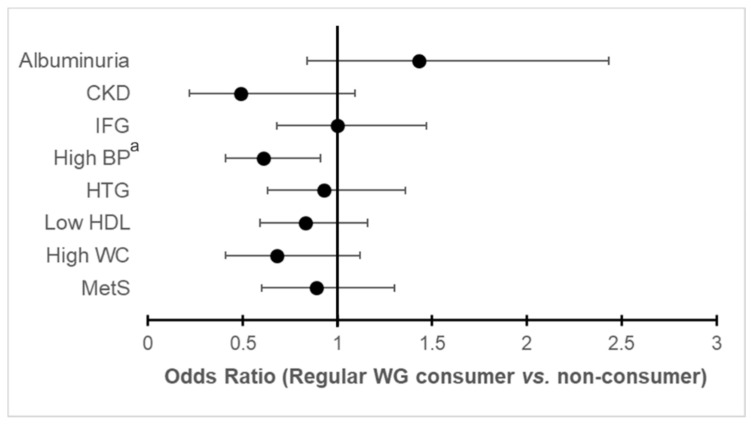
Odds ratio for regular wholegrain consumers (*n*: 3,124,834) vs. non-consumers (*n*: 6,369,362) on cardiorenal metabolic outcomes. MetS, metabolic syndrome; WC, waist circumference; HDL, high density lipoprotein; HTG, hypertriglyceridemia; BP, blood pressure; IFG, impaired fasting glucose; CKD, chronic kidney disease. *^a^ p* < 0.05. Model was adjusted for geographical area, age, sex, BMI, education level, tobacco use, frequency of alcohol consumption and quantity of glasses of alcohol typically consumed, physical activity, diabetes, previous cardiovascular disease and consumption of fruits, vegetables and legumes. For renal outcomes (CKD and albuminuria) the model also included hypertension. Albuminuria was available in 2,159,174 regular WG consumers (70%) and in 4,499,396 non-consumers (71%) and ordinal regression instead of logistic regression was used.

**Table 1 nutrients-12-02815-t001:** General characteristics of the participants according to the frequency of wholegrain (WG) consumption.

	All	Non–Consumers	Sporadic WG Consumers	Regular WG Consumers	*p* for Trend *
Interviewed participants (*n*)	3110	1739	613	758	
Weighted population (*n*,%)	11,810,647 (100)	6,369,362 (54)	2,316,451 (20)	3,124,834 (26)	
Age (years)	44.2 (0.5)	47.1 (0.8)	42.2 (1.1)	40.0 (0.9)	<0.001
Urban residents (%)	88.9 (0.7)	86.0 (1.1)	90.1 (1.5)	94.1 (1.1)	<0.001
Female sex (%)	51.5 (1.4)	48.3 (1.8)	54.9 (3.3)	55.5 (3.5)	0.025
**Education level (%)**					<0.001
<8 years	17.1 (1.1)	22.6 (1.7)	10.3 (1.5)	11.0 (2.0)	
8–12 years	54.5 (1.9)	58.9 (2.3)	54.8 (3.3)	45.4 (3.4)	
>12 years	28.4 (1.8)	18.5 (1.9)	35.0 (3.4)	43.6 (3.6)	
**BMI category (%)**					0.320
<25 kg/m^2^	22.5 (1.4)	21.2 (1.7)	19.8 (2.7)	27.0 (3.0)	
25–30 kg/m^2^	42.7 (1.8)	42.5 (2.3)	46.2 (3.5)	40.5 (3.2)	
>30 kg/m^2^	34.8 (1.5)	36.3 (2.0)	33.9 (3.1)	32.5 (3.1)	
**PA (%)**					0.044
Low	34.7 (1.5)	37.3 (2.1)	31.4 (3.2)	32.0 (3.0)	
Moderate	23.8 (1.4)	24.5 (1.8)	23.8 (2.8)	22.2 (2.8)	
High	41.5 (1.8)	38.2 (2.3)	44.8 (3.5)	45.7 (3.4)	
Tobacco use (%)	36.1 (1.5)	38.3 (2.0)	39.6 (3.3)	28.9 (3.2)	0.002
**Frequency of alcohol**					0.125
**consumption (%)**					
Never	27.1 (1.3)	25.4 (1.5)	25.4 (2.7)	31.6 (3.0)	
≤1 time/month	37.6 (1.5)	39.0 (2.0)	34.1 (3.2)	37.1 (3.0)	
2–4 times/month	24.8 (1.4)	25.2 (2.0)	29.6 (3.4)	20.4 (2.3)	
2–3 times/week	7.9 (1.1)	7.5 (1.4)	9.1 (2.2)	8.0 (2.0)	
>3 times/week	2.6 (0.5)	2.9 (0.7)	1.8 (0.8)	2.8 (1.1)	
Alcohol consumption (glasses)	2.2 (0.1)	2.4 (0.1)	2.2 (0.1)	1.9 (0.1)	<0.001
Fruits (times/week)	4.3 (0.1)	4..0 (0.1)	4.4 (0.1)	5.1 (0.2)	<0.001
Vegetables (times/week)	5.5 (0.1)	5.4 (0.1)	5.6 (0.1)	5.8 (0.1)	0.008
Legumes (%)					0.503
≤1 time/month	12.1 (1.1)	11.7 (1.3)	12.4 (2.4)	12.5 (2.4)	
>1 time/month	62.8 (1.8)	62.8 (2.3)	65.9 (3.3)	60.7 (3.1)	
≥2 times/week	25.1 (1.4)	25.5 (1.8)	21.7 (2.8)	26.8 (2.9)	
HT (%)	27.1 (1.3)	31.5 (1.9)	27.9 (3.1)	17.7 (2.1)	0.164
Diabetes (%)	12.8 (1.0)	13.0 (1.4)	12.4 (1.8)	12.6 (2.1)	0.368
CVD (%)	9.6 (0.9)	11.0 (1.3)	9.4 (1.8)	6.8 (1.2)	0.396
MetS (%)	41.7 (1.6)	45.2 (2.1)	42.3 (3.3)	34.1 (3.3)	0.323
CKD (%)	2.9 (0.5)	3.6 (0.7)	3.5 (1.1)	1.1 (0.3)	0.181
**uACR (%)**					0.962
0.34–3.39 mg/mmol	8.3 (0.8)	7.4 (0.9)	10.1 (2.0)	8.7 (2.1)	
>3.39 mg/mmol	1.4 (0.3)	1.7 (0.5)	1.5 (0.7)	0.9 (0.4)	

BMI, body mass index; PA, physical activity; HT, hypertension; CVD, cardiovascular disease; MetS, metabolic syndrome; CKD, chronic kidney disease; uACR, urinary albumin-creatinine ratio. * Generalized linear models adjusted for age and sex. Available data in uACR (2432 interviewed participants representing a population of 8,385,402).

**Table 2 nutrients-12-02815-t002:** Risk factors for cardiorenal metabolic diseases according to the frequency of WG consumption.

	Non–Consumers (*n*: 6,369,362) ^†^	Sporadic WG Consumers (*n*: 2,316,451) ^†^	Regular WG Consumers (*n*: 3,124,834) ^†^	*p* for Trend *
WC (cm)	94.5 (0.5)	93.6 (0.9)	91.7 (0.8)	0.156
SBP (mmHg)	126.1 (0.8)	124.1 (1.4)	118.7 (0.9)	0.006
DBP (mmHg)	75.6 (0.4)	74.9 (0.7)	72.2 (0.6)	0.007
Glucose (mmol/L)	5.36 (0.04)	5.31 (0.07)	5.11 (0.05)	0.389
HbA1c (mmol/mol)	43.2 (0.8)	45.9 (1.8)	41.6 (1.2)	0.993
TG (mmol/L)	1.72 (0.05)	1.60 (0.09)	1.48 (0.07)	0.007
TC (mmol/L)	4.67 (0.04)	4.66 (0.07)	4.50 (0.06)	0.319
HDL-C (mmol/L)	1.18 (0.01)	1.24 (0.02)	1.25 (0.02)	0.005
LDL-C (mmol/L)	2.70 (0.03)	2.69 (0.06)	2.59 (0.05)	0.459
VLDL-C (mmol/L)	0.78 (0.02)	0.71 (0.04)	0.66 (0.25)	0.008
Non-HDL-C (mmol/L)	3.50 (0.04)	3.42 (0.07)	3.25 (0.06)	0.071
TG/HDL-C	1.7 (0.07)	1.5 (0.1)	1.4 (0.1)	0.003
Creatinine (μmol/L)	70.7 (0.8)	69.5 (1.1)	68.4 (1.1)	0.251
GFR (mL/min.1.73 m^2^)	101 (1)	104 (1)	106 (1)	0.343
uACR (mg/mmol)	0.28 (0.05)	0.26 (0.04)	0.21 (0.04)	0.990

WC, waist circumference; SBP, systolic blood pressure; DBP, diastolic blood pressure; HbA1c, hemoglobin A1c; TG, triglycerides; TC, total cholesterol; HDL, high-density lipoprotein; LDL, low-density lipoprotein; VLDL, very low-density lipoprotein; GFR, glomerular filtration rate; uACR, urinary albumin-creatinine ratio. Values are mean (SE). ***** Generalized linear models adjusted for age and sex. Available data in HbA1c (1106 interviewed participants representing a population of 3,335,061) and uACR (2432 interviewed participants representing a population of 8,385,402). ^†^
*n* after applying sampling weights.

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
