# Peer review of "Wholegrain Consumption and Risk Factors for Cardiorenal Metabolic Diseases in Chile: A Cross-Sectional Analysis of 2016–2017 Health National Survey"

_nutrients, 2020, doi:10.3390/nu12092815_

Round 1

Reviewer 1 Report

The authors evaluated the association between whole grain consumption and cardiorenal metabolic diseases using the 2016-2017 Chile Health National Survey. Although the topic is interesting, several key points are needed to be revised or clarified.

  1. Line 62. The authors mentioned a healthy diet. Please describe a healthy diet among the Latin-American population in detail.
  2. Authors have focused on the whole grain in a Latin-American population with relatively low consumption compared to other populations such as Asian or European. Why did the authors concentrate on the consumption of whole-grain than those of other healthy food such as fruit and vegetable or low-fat and sodium food with regard to non-communicable disease? Please reinforce them in the introduction session.
  3. Please present the no. of IRB.
  4. In the current study, 2905 subjects (about 47%) exclude due to missing data on MetS and covariates. It needs to separate the criteria for MetS (main outcome) and confounding variables (covariate). And I suggest re-analysis that handled missing value for missing data of confounding variables (saying that again, including covariate). In the current analysis, the final sample size was half of the target population.
  5. Did the authors use the validated food frequency questionnaire? Please explain the information dietary assessment tool in more detail.
  6. Did authors apply strata, clusters, and weight to the statistical models based on complex sampling design (stratified multistage sampling) of the Chilean National Health Survey? If authors have used it, please mention it in statistical analysis.
  7. Authors adjusted for a geographical area, age, sex, BMI, education level, tobacco use, frequency of alcohol consumption and quantity of alcohol glasses typically consumed, physical activity, diabetes, previous cardiovascular disease and consumption of fruits, vegetables and legumes (and hypertension) in a multivariate model. How did you select them? If you have references or related to previous studies, please describe them.

Author Response

Reviewer 1:

The authors evaluated the association between whole grain consumption and cardiorenal metabolic diseases using the 2016-2017 Chile Health National Survey. Although the topic is interesting, several key points are needed to be revised or clarified.

  1. Line 62. The authors mentioned a healthy diet. Please describe a healthy diet among the Latin-American population in detail.

The Food Based Dietary Guidelines (FBDG) presents 11 key messages [1]. Although no message directly alludes to whole grains, it is recommended to avoid sugars and sweet products. The current graphic representation of FBDG is composed by a circle which represents variety and proportionality of food groups that should be consumed for a healthy diet. The circle includes 6 main food groups: fruits, vegetables, grains, protein foods, dairy, and oils. The foods included in the black bottom strip represent energy dense and low nutritional value foods which should be avoided [2,3]. Similar guidelines for food groups and graphical representations are available for other Latin American countries [4]. However, there is an important difference between the current intake of the population and the dietary recommendations all around the Region, including Chile [5].

A description of a healthy diet based on the Chilean FDBG has been added to the revised manuscript in the Introduction Section, lines 62-65

  1. Authors have focused on the whole grain in a Latin-American population with relatively low consumption compared to other populations such as Asian or European. Why did the authors concentrate on the consumption of whole-grain than those of other healthy food such as fruit and vegetable or low-fat and sodium food with regard to non-communicable disease? Please reinforce them in the introduction session.

The health benefits of whole grain (WG) on the risk for non-communicable diseases have been consistent among several studies and meta-analyses [6,7]. Moreover, a recent meta-analysis observed an inverse linear dose-response relationship between WG consumption and the risk for cardiorenal metabolic diseases [6]. Thus, our aim was to analyze the association between WG in a Latin-American population, characterized by low intake levels of WG foods, and the risk factors of cardiorenal metabolic disease. A recently published study already addressed the relationships between higher consumption of fruits/vegetables and of fish/seafood and the metabolic syndrome risk in the 2016–2017 Chilean National Health Survey [8]. In particular, in their analysis, the authors did not assess WG consumption. One of the possible reasons could be that the Chilean FBDG [1] does not make a recommendation for WG foods among their 11 key messages. Another possibility could be that WG foods were disregarded due to its low consumption in the Chilean population. Thus, our study highlights the importance of promoting WG consumption and provides evidence to consider its inclusion in the national dietary guidelines.

In the introduction section, this issue was developed further in lines 64-73.

  1. Please present the no. of IRB.

We added the IRB number 16-019 of the 2016-2017 Chilean National Health Survey (CNHS). Methods section, line 85.

  1. In the current study, 2905 subjects (about 47%) exclude due to missing data on MetS and covariates. It needs to separate the criteria for MetS (main outcome) and confounding variables (covariate). And I suggest re-analysis that handled missing value for missing data of confounding variables (saying that again, including covariate). In the current analysis, the final sample size was half of the target population.

The flowchart of participants showing how many subjects had missing data in the outcome and in any of the covariates is now included in the Figure 1 of the revised manuscript. In the 2016-2017 CNHS, blood and urine samples were taken from a stratified, random subsample [9]. For this reason, 47% of the subjects had missing values on MetS.

In the revised version of the manuscript we applied the Sampling weights from the Survey that accounted for differences in the selection probability, for non-response rates and post-stratification adjustment. Thus, the results are now expanded to an estimated total [95% CI] of 11,810,647 (10,982,016 - 12,639,278) individuals in Chile.

  1. Did the authors use the validated food frequency questionnaire? Please explain the information dietary assessment tool in more detail.

Dietary assessment was carried out by trained interviewers collecting the frequency of consumption of fish and seafood, dairy products, WG products, fruits, vegetables, legumes, sugar-sweetened beverages and the usual culinary fat. Therefore, the Survey was not a food frequency questionnaire. However, these 14 general questions with conditional issues were designed by experts based on the Food Based Dietary Guideline (FBDG) for the Chilean population and the experience of both previous Chilean National Health Surveys in 2003 and 2009-2010.

These questions asked how often the participants consumed 3 key food groups within the Mediterranean diet and which was the culinary fat used. Fruits and vegetables (times/week) and fish and seafood (four options: <1 time/month, 1 to <3 times/month, 4 times/month, or >4 times/month). In regards to dairy products, it asked two questions, one about the frequency of consumption (7 options, >3 times/day, 1-3 times/day, 1 time/day, 0.5 times/day, <1/week, <1/month or never) and the other about the type: (semi-)skimmed vs. whole. Other questions were regarding consumption of sweetened drinks and juices, water and awareness of nutritional labels. The remaining questions had already been described in the manuscript regarding the consumption of legumes and WG foods.

In the manuscript from lines 95-108 the description of the exact questions from the dietary assessment tool used in the CNHS had already been described. The other questions regarding variables not included in the analysis were not described in the manuscript as they were not used in the present analysis (culinary fat, dairy products, labeling, water and sweetened drinks, etc).

“WG consumption was classified in three categories according to the question: How often do you consume any WG cereal, like WG bread, WG cereal or any other food that contains WG flour? Regular WG consumers were the participants who answered “more than once a day”, “once a day” or “every two days”. Sporadic WG consumers were those who answered “at least once a week” or “once a month”. Non-consumers were those that reported to “never consume WG foods”.

The consumption of fruits, vegetables and legumes were selected as dietary covariates because they are also important sources of dietary fiber. Consumption of fruits and vegetables was estimated through the questions: “Typically, how many days a week do you eat fruits?” and “Typically, how many days a week do you eat vegetables or vegetable salad? (do not include legumes or potatoes)”, respectively. The answers in days per week were used as continuous variables. Consumption of legumes was assessed through the question: “How often do you consume any type of legumes, such as beans, lentils, green peas or chickpeas?” They were grouped into three categories: i)“two or more times per week”, ii) between “at least once a week” and “between 1 and 3 times a month”, and iii) “less than once a month or never”

  1. Did authors apply strata, clusters, and weight to the statistical models based on complex sampling design (stratified multistage sampling) of the Chilean National Health Survey? If authors have used it, please mention it in statistical analysis.

We would like to thank the reviewer for his/her comment that makes a major contribution to the manuscript. We hadn’t considered the complex sampling design of the Survey. The section of Statistical Analysis and Results, as well as the discussion, have been updated. The results are now based in 11,810,647 participants.

Extensive editing to fit the updated results has been made throughout the entire manuscript.

  1. Authors adjusted for a geographical area, age, sex, BMI, education level, tobacco use, frequency of alcohol consumption and quantity of alcohol glasses typically consumed, physical activity, diabetes, previous cardiovascular disease and consumption of fruits, vegetables and legumes (and hypertension) in a multivariate model. How did you select them? If you have references or related to previous studies, please describe them.

The selection of confounders was done a priori and was based on the associations with MetS in previous analyses of the Chilean NHS [8,10,11]. Confounders were also selected based on existing meta-analysis and in a physiological basis [12–15] In addition, further adjustment for other sources of dietary fiber, such as fruits vegetables and legumes, was done because of the benefits of wholegrain foods are mainly attributed to its fiber content. Lines 136-137 are now stating the selection criteria for the covariates of the study.

References

[1]       Olivares C. S, Zacarías H. I, González G. CG, Villalobos V E. Proceso de formulación y validación de las guías alimentarias para la población chilena. Rev Chil Nutr 2013;40:262–8. https://doi.org/10.4067/S0717-75182013000300008.

[2]       Cortés SO, Hasbún IZ, González CGG, Morán LF, Stoltze FM, Fernandes ACP, et al. Diseño y validación de la imagen para la difusión e implementación de las guías alimentarias para la población chilena. Nutr Hosp 2015;32:582–9. https://doi.org/10.3305/nh.2015.32.2.9084.

[3]       Food and Agriculture Organization of the United Nations (FAO). Food-based dietary guidelines - Chile n.d. http://www.fao.org/nutrition/education/food-dietary-guidelines/regions/countries/Chile (accessed September 3, 2020).

[4]       Betzaida M, Martínez A, Yanet Cordero Muñoz A, Ojeda GM, Fabiola Y, Sandoval M, et al. A review of graphical representations used in the dietary guidelines of selected countries in the Americas, Europe and Asia. Nutr Hosp 2015;32:986–96. https://doi.org/10.3305/nh.2015.32.3.9362.

[5]       Kovalskys I, Zonis L, Guajardo V, Rigotti A, Koletzko B, Fisberg M, et al. Latin American consumption of major food groups: Results from the ELANS study. PLoS One 2019;14. https://doi.org/10.1371/journal.pone.0225101.

[6]       Reynolds A, Mann J, Cummings J, Winter N, Mete E, Te Morenga L. Carbohydrate quality and human health: a series of systematic reviews and meta-analyses. Lancet 2019;393:434–45. https://doi.org/10.1016/S0140-6736(18)31809-9.

[7]       Micha R, Shulkin ML, Peñalvo JL, Khatibzadeh S, Singh GM, Rao M, et al. Etiologic effects and optimal intakes of foods and nutrients for risk of cardiovascular diseases and diabetes: Systematic reviews and meta-analyses from the nutrition and chronic diseases expert group (NutriCoDE). PLoS One 2017;12. https://doi.org/10.1371/journal.pone.0175149.

[8]       Fernández-Verdejo R, Moya-Osorio JL, Fuentes-López E, Galgani JE. Metabolic health and its association with lifestyle habits according to nutritional status in Chile: A cross-sectional study from the National Health Survey 2016-2017. PLoS One 2020;15:e0236451. https://doi.org/10.1371/journal.pone.0236451.

[9]       Ministerio de Salud, Gobierno de Chile. Encuesta Nacional de Salud 2016–2017 . Online: Http://EpiMinsalCl/Encuesta-Nacional-de-Salud-2015-2016/ 2018:(Last accessed on May 2020).

[10]    Valenzuela Andrea A B, Maíz A, Margozzini P, Ferreccio C, Rigotti A, Olea R, et al. Prevalencia de síndrome metabólico en población adulta chilena: Datos de la encuesta nacional de salud 2003. Rev Med Chil 2010;138:707–14. https://doi.org/10.4067/s0034-98872010000600007.

[11]    Sadarangani KP, Von Oetinger A, Cristi-Montero C, Cortínez-O’Ryan A, Aguilar-Farías N, Martínez-Gómez D. Beneficial association between active travel and metabolic syndrome in Latin-America: A cross-sectional analysis from the Chilean National Health Survey 2009–2010. Prev Med (Baltim) 2018;107:8–13. https://doi.org/10.1016/j.ypmed.2017.12.005.

[12]    Bellou V, Belbasis L, Tzoulaki I, Evangelou E. Risk factors for type 2 diabetes mellitus: An exposure-wide umbrella review of meta-analyses. PLoS One 2018;13. https://doi.org/10.1371/journal.pone.0194127.

[13]    Chen JP, Chen GC, Wang XP, Qin L, Bai Y. Dietary fiber and metabolic syndrome: A meta-analysis and review of related mechanisms. Nutrients 2018;10. https://doi.org/10.3390/nu10010024.

[14]    Ilanne-Parikka P, Eriksson JG, Lindström J, Hämäläinen H, Keinänen-Kiukaanniemi S, Laakso M, et al. Prevalence of the metabolic syndrome and its components: Findings from a Finnish general population sample and the Diabetes Prevention Study cohort. Diabetes Care 2004;27:2135–40. https://doi.org/10.2337/diacare.27.9.2135.

[15]    Gorter PM, Olijhoek JK, Van Der Graaf Y, Algra A, Rabelink TJ, Visseren FLJ. Prevalence of the metabolic syndrome in patients with coronary heart disease, cerebrovascular disease, peripheral arterial disease or abdominal aortic aneurysm. Atherosclerosis 2004;173:361–7. https://doi.org/10.1016/j.atherosclerosis.2003.12.033.

Reviewer 2 Report

WEll-written and well set out; well-defined

Interesting gender differences in WG consumption

Thre are a few mino changes that need to be addressed:

  1. p8 of 13: In consequence, the analyses awere repeated in men and women by separate...? There seems to be a word or phrase missing
  2. Table 2  (p7 of 13)
  3. HBA1c seems to be expressed in different units in the regular WG consumption group

Author Response

WEll-written and well set out; well-defined

Interesting gender differences in WG consumption

We thank the reviewer for his/her comments.

Thre are a few mino changes that need to be addressed:

  1. p8 of 13: In consequence, the analyses awere repeated in men and women by separate...? There seems to be a word or phrase missing

The sentence was rewritten for better understanding.

  1. Table 2  (p7 of 13). HBA1c seems to be expressed in different units in the regular WG consumption group.

We corrected the units of HbA1c in Table 2.

Round 2

Reviewer 1 Report

Thank you for your revisions, I do not have any other comments.